

# Preeclampsia in pregnant women with COVID-19: a prospective cohort study from two tertiary hospitals in Southern Brazil

Narcizo LEC Sobieray[1], Newton S. Carvalho[2], Cynthia F. Klas[3], Isabella N. Furuie[3], Jullie A. Chiste[3], Cyllian A. Fugaça[3], Jessica S. Longo[3], Juliana D. Oliveira[4] and Sérgio L. Padilha[5]

[1] Department of Obstetrics and Gynecology and Postgraduate Program in Internal Medicine and Health Sciences, Universidade Federal do Paraná, Curitiba, Paraná, Brazil

[2] Department of Obstetrics and Gynecology and Postgraduate Program in Obstetrics and Gynecology, Universidade Federal do Paraná, Curitiba, Paraná, Brazil

[3] Department of Obstetrics and Gynecology, Hospital of Clinics Complex—CHC-UFPR-EBSERH, Universidade Federal do Paraná, Curitiba, Paraná, Brazil

[4] Department of Obstetrics and Gynecology, Hospital Nossa Senhora das Graças, Curitiba, Paraná, Brazil

[5] Department of Internal Medicine and Postgraduate Program in Internal Medicine and Helath Sciences, Universidade Federal do Paraná, Curitiba, Paraná, Brazil

Corresponding author
Narcizo LEC Sobieray,
narsobieray@gmail.com

## ABSTRACT

**Background.** COVID-19 is an infectious pathology that shows vascular changes during pregnancy, as well as in the placentas. The main objectives of this study were to estimate the prevalence and the risk factors for preeclampsia in hospitalized pregnant women with COVID-19. As well as comparing maternal and perinatal outcomes in hospitalized pregnant women with COVID-19 and preeclampsia with those without preeclampsia.

**Methods.** Prospective cohort study of 100 hospitalized pregnant women from two tertiary hospitals, diagnosed with COVID-19, and divided into two groups: PE+ group (pregnant women with COVID-19 and preeclampsia) and PE- group (pregnant women with COVID-19 without preeclampsia). These pregnant women had prevalence, risk factors, maternal and perinatal data analyzed.

**Results.** The prevalence of preeclampsia was 11%. Severe COVID-19 was the main risk factor for preeclampsia (OR = 8.18 [CI 1.53–43.52]), as well as fetal growth restriction was the main perinatal outcome (OR = 8.90 [CI 1.52–38.4]). Comorbidities were more frequent in the PE+ group (63.6% *vs* 31.5%, $p = 0.03$), as well as prematurity (81.8% *vs* 41.6%, $p = 0.02$), low birth weight (63.6% *vs* 24.7%, $p = 0.01$), and the need for neonatal intensive care admission of the newborn (63.6% *vs* 27.0%, $p = 0.03$). Pregnant women with PE had twice as long a length of stay in the intensive care unit (RR = 2.35 [CI 1.34–4.14]). Although maternal mortality was more frequent among pregnant women with PE, it was not statistically significant.

**Conclusions.** Prevalence of preeclampsia in hospitalized pregnant women with COVID-19 was 11%. Severe COVID-19 was the main risk factor for preeclampsia and associated comorbidities increased the risk for developing preeclampsia. Long length of stay in the intensive care unit was the main maternal outcome and fetal growth restriction was the main perinatal outcome of preeclampsia.

# INTRODUCTION

COVID-19 a pathology caused by a new virus SARS-CoV-2, is an acute respiratory infection with high transmissibility and potentially serious, has been the most critical public health crisis to occur in this century (*Karimi-Zarchi et al., 2021*). Despite being primarily a respiratory disease, it has important systemic effects such as high blood pressure, thrombocytopenia and impact on organs such as the liver, kidneys, thrombotic system, placentas as well as in brain (with increased risk and severity of neurodegenerative disorders) (*Bedran, Bedran & Kote, 2024*). COVID-19 is also associated with immune activation that results in elevated levels of pro-inflammatory cytokines, including interleukin (IL): IL-2, IL-6, IL-7, and tumor necrosis factor-α (TNF-α) (*Karimi-Zarchi et al., 2021*). Pregnant women with severe COVID-19 had worse clinical outcomes than non-pregnant women, including increased risk for admission to the Intensive Care Unit (ICU), use of invasive mechanical ventilation, need for extracorporeal membrane oxygenation, and death (*Lai et al., 2021*). Therefore, pregnant women are considered a risk group for severity of COVID-19. In addition, SARS-CoV-2 infection during pregnancy is a risk factor for preeclampsia (PE), fetal death, and preterm birth (*Lai et al., 2021*; *Metz et al., 2021*).

PE is a multifactorial, multisystemic, pregnancy-specific hypertensive disease of unknown etiology (*Poon et al., 2019*). It is considered a clinical manifestation of a disease of the maternal vascular endothelium, which is mediated by the placenta (*Poon et al., 2019*). Also, it is a result of inappropriate trophoblastic invasion of the spiral arterioles of the uterus, leads to an hypoxic environment, oxidative stress and, local and systemic inflammation (*Tossetta et al., 2023*). All these are characteristics of PE pregnancies. PE manifests itself after the 20th week of gestation in pregnant women who were previously normotensive (*Poon et al., 2019*). As pregnancy progresses, PE can lead to more severe situations, such as eclampsia, hemorrhagic stroke, HELLP syndrome (hemolysis, elevated liver enzymes, and low platelet count), renal failure, acute pulmonary edema, and death (*Weinstein, 1982*).

The prevalence of PE ranges in pregnancy varies from 2 to 5% worldwide (*Lisonkova et al., 2014*). In Brazil, the prevalence of eclampsia varies according to the region of the country. It ranges from 0.2% in the more developed regions to 8.1% in the less developed ones, thus making it the leading cause of maternal morbidity and mortality in these regions (*Giordano et al., 2014*).

SARS-CoV-2 infection can cause endothelial dysfunction, systemic intravascular inflammation, proteinuria, thrombin activation (microthrombi deposition), microvascular dysfunction and high blood pressure, all of which are key features of the pathophysiology of PE (*Karimi-Zarchi et al., 2021*; *Sánchez-Aranguren et al., 2014*; *Hantoushzadeh et al., 2022*; *Dap & Morel, 2020*). Inflammatory cytokines activate B lymphocytes and increase the production of autoantibodies to angiotensin 2 receptor type 1, which in turn stimulates

the increase of endothelin-1 and soluble sFlt-1-tyrosine kinase expression (*Herrock et al., 2022*; *Lamarca, Wallace & Granger, 2011*). In addition, there is increased oxidative stress, vasoactive imbalance, vasoconstriction, placental poor perfusion, platelet activation, and coagulation cascade (*Sánchez-Aranguren et al., 2014*; *Sanchez et al., 2021*). During the COVID-19 pandemic, the aims of this study were to estimate the prevalence of PE and to identify the main predictive factors for PE among hospitalized pregnant women with COVID-19. As well as comparing maternal and perinatal outcomes in pregnant women with COVID-19 and PE with those without PE.

## METHODS

The research consists in an observational, prospective study with cohort design of 100 hospitalized pregnant women diagnosed with COVID-19 and divided into two groups: PE+ (with PE) and PE- (without PE). The study was conducted from July 2020 to July 2021, in two tertiary hospitals from Curitiba, in southern Brazil. The study was approved by the Human Research Ethics Committee of the Hospital de Clínicas Complex of the Federal University of Paraná on CAAE number 35129820.6.0000.0096 dated June 30, 2020. The Free and Informed Consent Form was applied by the researchers and was obtained in writing from each participant.

**Inclusion criteria:** hospitalized pregnant women with positive tests for SARS-CoV-2, through nasopharyngeal real time polymerase chain reaction (RT-PCR), rapid antigen test, and IgM and IgG serology; with gestational age (GA) greater than 20 weeks; regardless of maternal age; presence of comorbidities; sociodemographic profile, and within the study period. Group of pregnant women with PE (PE+): pregnant women with blood pressure (BP) $\geq$ 140/90 mmHg, proteinuria/creatinuria ratio $\geq$ 30 mg/mol (*Poon et al., 2019*). US-Doppler with uterine artery pulsatility index (UtAPI) >95th percentile as a predictive factor for PE. Cases diagnosed with HELLP syndrome. Group of pregnant women without PE (PE-): pregnant women who do not meet these proposed diagnostic criteria for PE. **Exclusion criteria:** unconfirmed suspected cases, pregnant women transferred to other institutions, SARS-CoV-2 acquired before pregnancy, GA before 20 weeks, multiple pregnancies, pregnant women with PE-like syndrome and HELLP-like syndrome associated with severe COVID-19, to avoid confounding factors with classic HELLP syndrome. However, it was temporary and showed improvement after the regression of viral pneumonia.

During the research, the data collection tools used were the software Tasy, the management application for University hospitals, and the hospital information system. Microsoft Excel spreadsheets were used to record the data. The variables studied included: (a) **Gestational history**: age of the pregnant woman (risk age greater than 34 years: absent or present), parity, usual or high-risk prenatal care; as main comorbidities studied: chronic arterial hypertension, obesity and gestational diabetes. (b) **Characteristics of severe COVID-19**: RT-PCR positive tests for COVID-19, clinical features of severe acute respiratory syndrome (SARS) such as flu-like syndrome that have dyspnea/respiratory discomfort or persistent pressure in the chest or O2 saturation <95% in room air,

respiratory frequency <10 bpm or >24 bpm, heart rate <49 or >119 bpm, temperature (°C) <35 or >37.8, any change in level of consciousness or bluish discoloration of lips or face (pregnant women: important to observe hypotension or high blood pressure and oliguria); chest axial computerized tomography (chest CAT, revealing ground glass image >50% of lungs); elevated inflammatory markers (PCR, ferritin, D-dimer); altered laboratory tests such as lymphopenia, proteinuria and elevated transaminases among others. (c) **Perinatal outcomes**: type of delivery, postpartum hemorrhage (PPH), premature rupture of membranes (PROM), histological placental changes, fetal death, prematurity, low birth weight, 5th minute Apgar score, need for hospitalization of the infant newborn in the Neonatal Intensive Care Unit (NICU). In the interpretation of perinatal outcomes, the criteria of the Brazilian Society of Pediatrics (*Ministério da Saúde Brazil, 2023*). (d) **Maternal outcomes**: ICU admission, duration of COVID-19 symptoms, length of stay in the ICU, need for ventilatory support, plasmapheresis, pronation, maternal corticosteroid therapy and maternal death. The classification of COVID-19 was standardized according to the Brazilian Ministry of Health's Manual of Recommendations for Assistance to Pregnant and Postpartum Women Facing the COVID-19 Pandemic (*Ministry of Health, 2023*).

## Statistical analysis

Student's $t$-test was used to estimate the difference between continuous variables with symmetric distribution and Mann–Whitney test was used for asymmetrical distribution. Fisher's exact test and Pearson's chi-square test were used for categorical variables. Relative risk (RR) was used for univariate analysis of risks associated with maternal outcomes. The multivariate logistic regression model was applied to estimate the main gestational and disease severity factors predictive for PE and main perinatal and maternal outcomes of PE. In the multivariate logistic regression model, the variable PE was considered as a dichotomous response variable. As a measure of effect, was estimated the relative risk (RR) for all variables studied and its 95% confidence interval (CI). The $p$ value $< 0.05$ is an indicator of statistical significance. The sample size was estimated at 93 cases, considering type II error of 10%, magnitude of effect of 10%, significance level of 5%, the prevalence of PE in Brazil of 2% and 12% in pregnant women with COVID-19, with 95% test power. The Statistica 4.0 software (SSPSv. 4.0, Palo Alto, CA, USA) was used for the analysis.

## RESULTS

Initially, 142 pregnant women with clinical suspicion of COVID-19, who were admitted from two tertiary hospital complexes, were studied. Of these, 42 were excluded by the criteria. Therefore 100 pregnant women with COVID-19 were included in the study, as shown in the flowchart for inclusion of pregnant women in the sample population. The PE+ group included 11 pregnant women and the PE- group included 89 pregnant women (Fig. 1).

Regarding gestational history, was observed no difference between PE+ and PE- groups with respect to maternal age (28.6 ± 8.0 *vs* 31.1 ± 6.3 years, $p = 0.23$). The average gestational age at delivery for the PE+ group was 33.4 + 3.4 weeks and in the PE- group it was 36.3 + 3.4 weeks ($p = 0.02$). Variables such as nulliparity and high risk prenatal

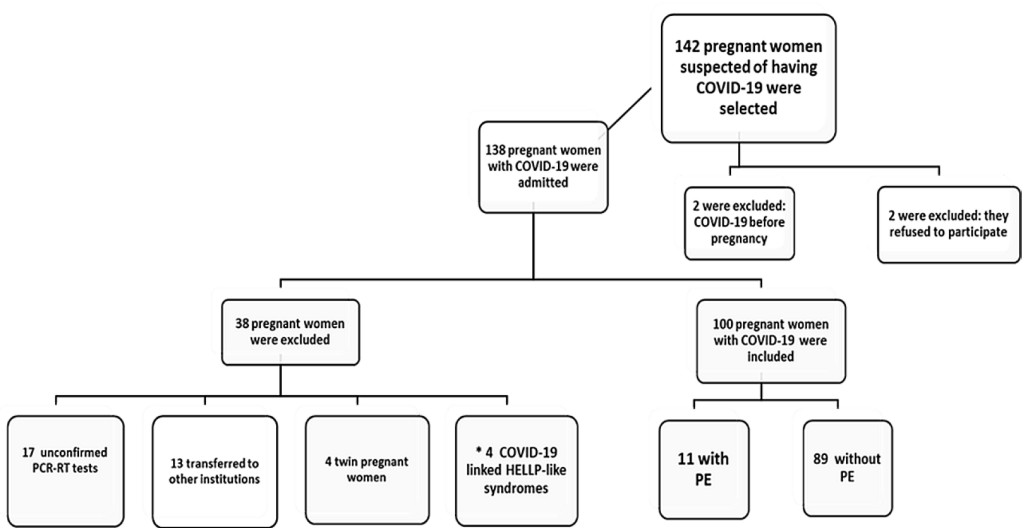

**Figure 1  Flowchart of pregnant women inclusion in the Sample Population.** Note: * "COVID-19 linked HELLP-like syndrome", possible temporary inflammatory syndrome with clinical and laboratory characteristics similar to classic HELLP syndrome, which were stabilized or showed clinical and laboratory improvement with normal UtAPI, were excluded by the criteria, to avoid confounding factors. Adapted: According to *Page et al. (2021)*.

were about 40% higher in PE+ group, not statistically significant, but the frequency of comorbidities was significantly higher in the PE+ group (63.6% *vs* 31.5%, $p = 0.03$). The main comorbidities of PE- group were diabetes and obesity, while diabetes, obesity, and hypertension were significantly higher in the PE+ group (Table 1).

The severity of COVID-19 was higher among pregnant women with PE, with a higher frequency of severe acute respiratory syndrome (SARS) (54.5% *vs* 24.7%, $p = 0.04$) (Table 1). Severe COVID-19 was the main predictor for PE, increasing its risk by eight times (OR = 8.18; CI [1.53–43.52]; $p < 0.01$) (Fig. 2).

Among the variables related to perinatal outcomes there was a higher frequency of fetal deaths in PE+ group (18.2% *vs* 2.2% $p = 0.05$). Prematurity rate was also higher in PE+ group (81.8% *vs* 41.6%, $p = 0.02$). As well as low birth weight (63.6% *vs* 24.7%, $p = 0.01$), NICU admission (63.6% *vs* 27.0%, $p = 0.03$) and FGR (27.3% *vs* 4.5%, $p = 0.02$), the latter being the main perinatal outcome in pregnant women with COVID-19 and PE (OR = 8.90; CI [1.52–38.4]; $p = 0.01$) (Fig. 3). The placental histological changes were demonstrated inflammatory histological changes (chronic histiocytic intervillositis), microvascular thrombosis, and perivillous fibrin deposition 30% higher in PE group (100% *vs* 74.6% $p = 0.10$) (Table 2).

Among maternal outcomes, the length of stay in the ICU was longer in the PE+ group ($p = 0.04$) which occurred on average at $9.7 \pm 2.5$ days of stay. No differences were observed in the frequency of maternal deaths ($p = 0.29$), one in the PE+ group and two in the PE- group. The frequency of need for ICU admission and pronation were also higher in PE+ group ($p = 0.06$ and $p = 0.08$) (Table 3).

**Table 1  Pregnancy characteristics and COVID-19 severity.**

| Pregnancy characteristics COVID-19 severity | PE+ (n = 11) mean ± SD/n (%) | PE- (n = 89) mean ± SD/n (%) | p value |
|---|---|---|---|
| Gestational age at time of delivery (weeks) | 33.4 + 3.4 | 36.3 + 3.4 | 0.02[a] |
| Age (years) | 28.6 ± 8.0 | 31.1 ± 6.3 | 0.23[a] |
| Risk age | 4 (36.4%) | 29 (32.6%) | 1.00[b] |
| Nulliparity | 9 (81.8%) | 53 (59.6%) | 0.19[b] |
| High risk prenatal care | 9 (81.8%) | 36 (40.4%) | 0.19[b] |
| Chronic hypertension | 3 (27.3%) | 10 (11.2%) | 0.15[b] |
| Obesity | 3 (27.3%) | 16 (18.0%) | 0.43[b] |
| Gestational Diabetes | 3 (27.3%) | 16 (18.0%) | 0.43[b] |
| PROM | 0 (0.0%) | 14 (15.7%) | 0.35[b] |
| Comorbidities | 7 (63.6%) | 28 (31.5%) | 0.03[b] |
| SARS | 6 (54.5%) | 22 (24.7%) | 0.04[b] |
| Severe COVID-19 | 9 (81.8%) | 30 (33.7%) | <0.01[b] |

**Notes.**

[a] Student's t test.

[b] Fisher's exact test.

SD, standard deviation; Comorbidities, chronic hypertension and/or obesity and/or gestational diabetes; PROM, premature rupture of membranes; PE+, pregnant women with COVID-19 with preeclampsia; PE-, pregnant women with COVID-19 without preeclampsia COVID-19 severity was characterized by the presence of severe acute respiratory syndrome (SARS). CAT of the chest with "ground-glass" imaging > 50% and elevated inflammatory markers.

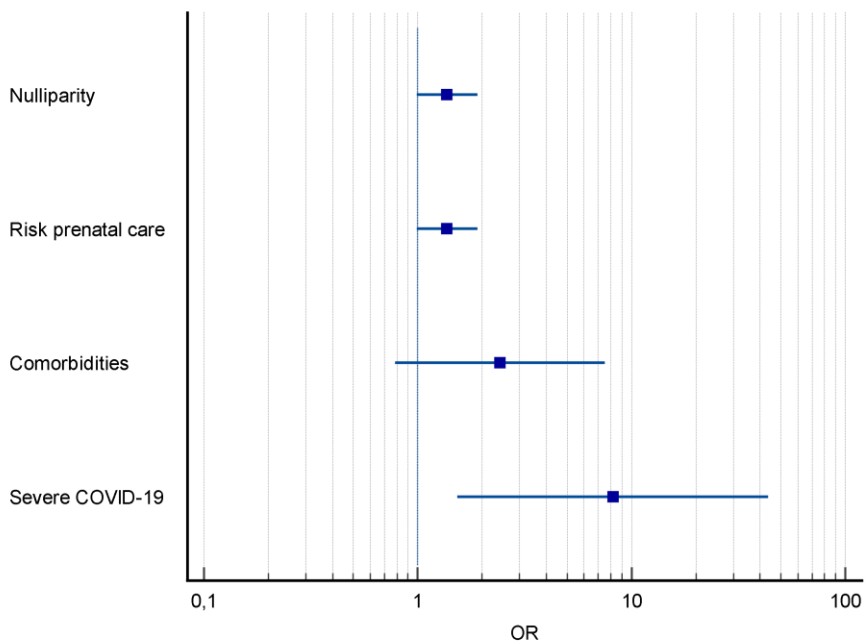

**Figure 2  Forest plot of pregnancy characteristics and COVID-19 and risk factors for preeclampsia.**
Notes: OR, odds ratio; CI, 95 percent confidence interval.

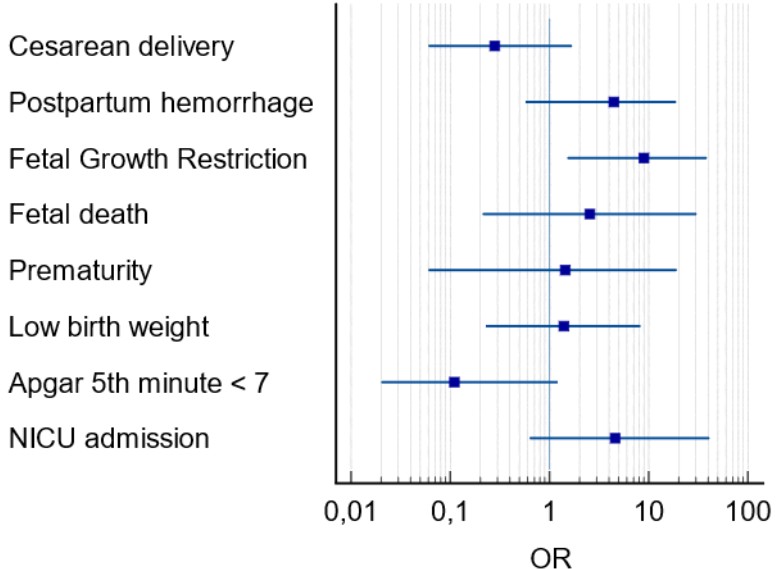

**Figure 3** Forest plot of perinatal outcomes in pregnant women with COVID-19 and preeclampsia.
Notes: OR, odds ratio; CI, 95 percent confidence interval; NICU, neonatal intensive care unit.

**Table 2  Perinatal outcomes in pregnant women with COVID-19.**

| Perinatal Outcomes | PE+ (*n* = 11) mean ± SD/n (%) | PE- (*n* = 89) mean ± SD/n (%) | *p* value |
|---|---|---|---|
| Acute COVID-19 at the time of delivery | 6 (54.5%) | 42 (47.2%) | 0.75[a] |
| Cesarean delivery | 8 (72.7%) | 68 (76.4%) | 0.72[a] |
| Postpartum hemorrhage | 2 (18.2%) | 5 (5.6%) | 0.17[a] |
| Fetal Growth Restriction | 3 (27.3%) | 4 (4.5%) | 0.02[a] |
| Fetal death | 2 (18.2%) | 2 (2.2%) | 0.05[a] |
| Placental Histological Changes[d] | 10 (100.0%) | 53 (74.6%) | 0.10[a] |
| Prematurity | 9 (81.8%) | 37 (41.6%) | 0.02[a] |
| Birth weight | 2120.4 ± 782.2 | 2851.1 ± 769.8 | <0.001[b] |
| Low birth weight | 7 (63.6%) | 22 (24.7%) | 0.01[a] |
| 5th minute Apgar score (medians and IIQ) | 9 (8-9) | 9 (9-10) | 0.37[c] |
| Apgar 5th minute < 7 | 2 (18.2%) | 7 (7.9%) | 0.25[a] |
| NICU admission | 7 (63.6%) | 24 (27.0%) | 0.03[a] |

Notes.
[a] Fisher's exact test.
[b] Student's t test.
[c] Mann–Whitney's test.
[d] *n* = 10.

SD, standard deviation; NICU, neonatal intensive care unit; PE+, pregnant women with COVID-19 with preeclampsia; PE-, pregnant women with COVID-19 without preeclampsia.

**Table 3  Maternal outcomes in pregnant women with COVID-19.**

| Maternal Outcomes | PE+ (*n* = 11) n (%)/median (IQR) | PE- (*n* = 89) n (%)/median (IQR) | *p* value |
|---|---|---|---|
| ICU admission | 6 (54.5%) | 22 (24.7%) | 0.06[a] |
| Time of symptoms (days) | 7 (3–11) | 5 (2–8) | 0.22[b] |
| ICU length of stay (days) > 7 | 5 (3–12) | 3 (2–6) | 0.04[b] |
| Mechanical ventilation | 4 (36.4%) | 18 (20.2%) | 0.22[a] |
| Plasmapheresis | 2 (18.2%) | 5 (5.6%) | 0.17[a] |
| Pronation | 4 (36.4%) | 13 (14.6%) | 0.08[a] |
| Maternal corticoid therapy | 5 (45.4%) | 25 (28.1%) | 0.29[a] |
| Time of corticoid therapy > 10 days | 4 (36.4%) | 25 (28.1%) | 0.17[a] |
| Maternal death | 1 (9.1%) | 2 (2.2%) | 0.29[a] |

Notes.
[a] Fisher's exact test.
[b] Mann–Whitney's test.
ICU, intensive care unit; PE+, pregnant women with COVID-19 with preeclampsia; PE-, pregnant women with COVID-19 without preeclampsia; IQR, interquartile range.

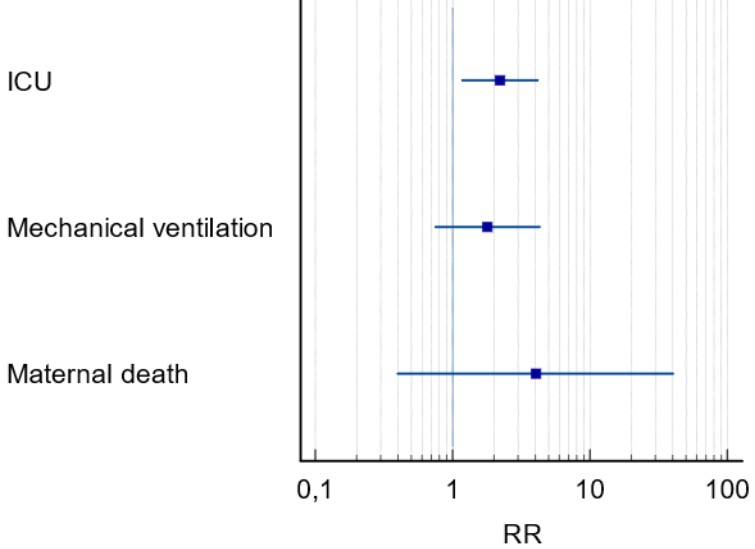

**Figure 4  Forest plot of maternal outcomes in pregnant women with COVID-19 and preeclampsia.**
Notes: RR, relative risk; CI, 95 percent confidence interval; ICU, intensive care unit.

In the multivariate analysis none of the variables were selected as predictive of the maternal outcomes of PE, but in the univariate analysis it indicated a two times higher risk for ICU admission and longer ICU stays in PE+ group (RR = 2.35; CI [1.34–4.14]; *p* = 0.04) (Fig. 4).

## DISCUSSION

Some authors consider COVID-19 in pregnancy as a risk factor for the development of PE, describing pathophysiological similarities at the endothelial level, with activation of the coagulation cascade in the most severe forms of the disease (*Lai et al., 2021*; *Sánchez-Aranguren et al., 2014*; *Hantoushzadeh et al., 2022*; *Dap & Morel, 2020*; *Sanchez et al., 2021*). In another American study, it was reinforced the difficulty of differential diagnosis between PE in pregnancies complicated by COVID-19 and transient inflammatory syndrome, which mimicked PE and HELLP syndrome. It recommended the use of specific molecular markers to rule out the diagnosis of PE. (*Jayaram et al., 2021*)

The prevalence of PE in hospitalized pregnant women with COVID-19 in this study was 11%, a similar result to the Brazilian multicenter study (REBRACO) (*Guida et al., 2022*) of 10%, and higher than those found by European authors (*Lai et al., 2021*; *Epelboin et al., 2021*). The difference between our study and the REBRACO study, is that prevalence of PE in REBRACO study was the same in the group with COVID-19 (10%) and in the group without COVID-19 (13%) (*Guida et al., 2022*). However, two studies, from Iran and from France suggested that symptomatic COVID-19 infection during pregnancy does not appear to increase the risk of PE and without increasing its prevalence (*Karimi-Zarchi et al., 2021*; *Tran et al., 2022*). The prevalence of PE in our study was considered high, three to four times higher than the prevalence observed in the population of pregnant women without COVID-19 from the same hospital institutions and like other historical series of COVID-19 on pregnancy (*Mendoza et al., 2020*; *Figueras et al., 2020*). Comparing the PE historical series from two participating institutions, we observed that both had a PE prevalence of 2.98% in the pre-pandemic period, therefore had a 3.7-fold increase in PE prevalence during the pandemic period. Furthermore, in the presence of severe disease, the PE prevalence in our study was three times higher, similar to studies from Spain (*Mendoza et al., 2020*; *Figueras et al., 2020*) and from the UK (*Lai et al., 2021*).

Regarding pregnancy characteristics, was observed a higher association with comorbidities in PE+ group, especially obesity and diabetes. The risk of FGR was eight times higher in PE+ group, being the main perinatal outcome, unlike a classic European study (*Rizzo et al., 2021*) that found no significant difference. The chance of developing PE was three times higher when there were associated comorbidities, while the Brazilian multicenter study showed an increase in PE when there was chronic hypertension or obesity separately (*Guida et al., 2022*). The main predictor of PE was the severe COVID-19, with an 8-fold increase in risk. This study showed high rates of cesarean births, around 73%, and no significant difference between groups, unlike the Brazilian multicenter study that had a significantly higher C-section rate in the group with PE (*Guida et al., 2022*).

In the literature, the possibility of the existence of a new temporary, inflammatory syndrome with clinical, laboratory, and pathophysiological features like the classic HELLP syndrome (*Sibai, 2004*; *Tou et al., 2022*; *Sobieray et al., 2023*) is currently described. This supposed new syndrome was named, as "COVID-19-linked HELLP-like syndrome" (CLHLS) and is associated with severe COVID-19 in pregnant women (*Sobieray et al.,*

*2023*), with a prevalence of 28.6% in these cases (*Mendoza et al., 2020*; *Figueras et al., 2020*).

In our study we found three cases of CLHLS in pregnant women with severe COVID-19 (3/29 –10.3%), and one case in a pregnant woman with mild COVID-19 (1/71 –1.4%). There are few studies available in the literature regarding mild and asymptomatic forms of COVID-19 that, in some cases, also caused the onset of PE (*Tossetta et al., 2022*) However, increased blood pressure found in COVID-19 pregnancies does not allow to associate COVID-19 to preeclampsia as hypertension is a common factor to both conditions (*Tossetta et al., 2022*). Some authors suggest the antiangiogenic/angiogenic (sFlt-1/PlGF) ratio, that when below 38 discards the diagnosis of PE and HELLP syndrome (*Tou et al., 2022*; *Zeisler et al., 2016*). Clinical, laboratory and UtAPI - Doppler data were analyzed by authors and overlap may have occurred in some cases.

The maternal death rate of 3% in the pregnant with COVID-19, was lower than the Brazilian rate of 7.2%, like the worldwide rates of 2.8% (*BRAZIL MH FIOCRUZ, 2021*). In Curitiba, South of Brazil, there was in 2021, an increase above three times in the maternal death rates, compared to the pre-pandemic period, half of them caused by COVID-19 and the maternal death rate was 0.2% (*BRAZIL MH FIOCRUZ, 2021*). Although maternal death showed no difference in the statistical analysis between two groups, there were limitations to this analysis, because another death occurred in a twin pregnant woman with PE, who was excluded by the criteria. The results showed 9.1% of maternal deaths in the PE+ group and 2.2% in the PE- group. In our study were related with higher severity of COVID-19 and increased the length of stay in the ICU in the PE+ group. The length of stay in the ICU was twice as long in the PE+ group, like another Brazilian study (*Guida et al., 2022*). There was no difference in the need for invasive mechanical ventilation, despite a 3-fold increase in the prevalence of PE in the severe COVID-19 cases, similar to the UK study (*Lai et al., 2021*).

As for the perinatal outcomes, demonstrated a 1.5-fold increase in prematurity in PE+ group (81.8% × 41.6%), similar to the results of the UK study (*Lai et al., 2021*) and the Brazilian study (*Guida et al., 2022*). Regarding prematurity rates in Brazil up to 11% in the pre-pandemic period and in the world, it was between 5 and 18% (*Ministério da Saúde Brazil, 2023*; *WHO, 2022*), there was an increase of up to 6 times within the groups. Low birth weight, NICU admissions were significantly higher in the PE+ group, similar to Brazilian study (*Guida et al., 2022*). The placental morphological findings demonstrated chronic histiocytic intervillositis, microvascular thrombosis, and perivillous fibrin deposition 30% higher in the PE+ group. In addition to poor placental perfusion and areas of infarcts, typical of PE. Therefore, the most important and practical information that can be extracted from this study is that when we are faced with a pregnant woman with COVID-19, the chance of association with PE is greater. If COVID-19 is severe or the pregnant woman shows comorbidities this chance increases to much. On the other hand, when in the pregnant woman with COVID-19 and the PE is associated, the chance of the severity of the case and the necessity of ICU are increased (Fig. 5).

Limitations include the non-availability of tests to evaluate biomarkers in the laboratories of the participating institutions, for differential diagnosis between HELLP-like and classic

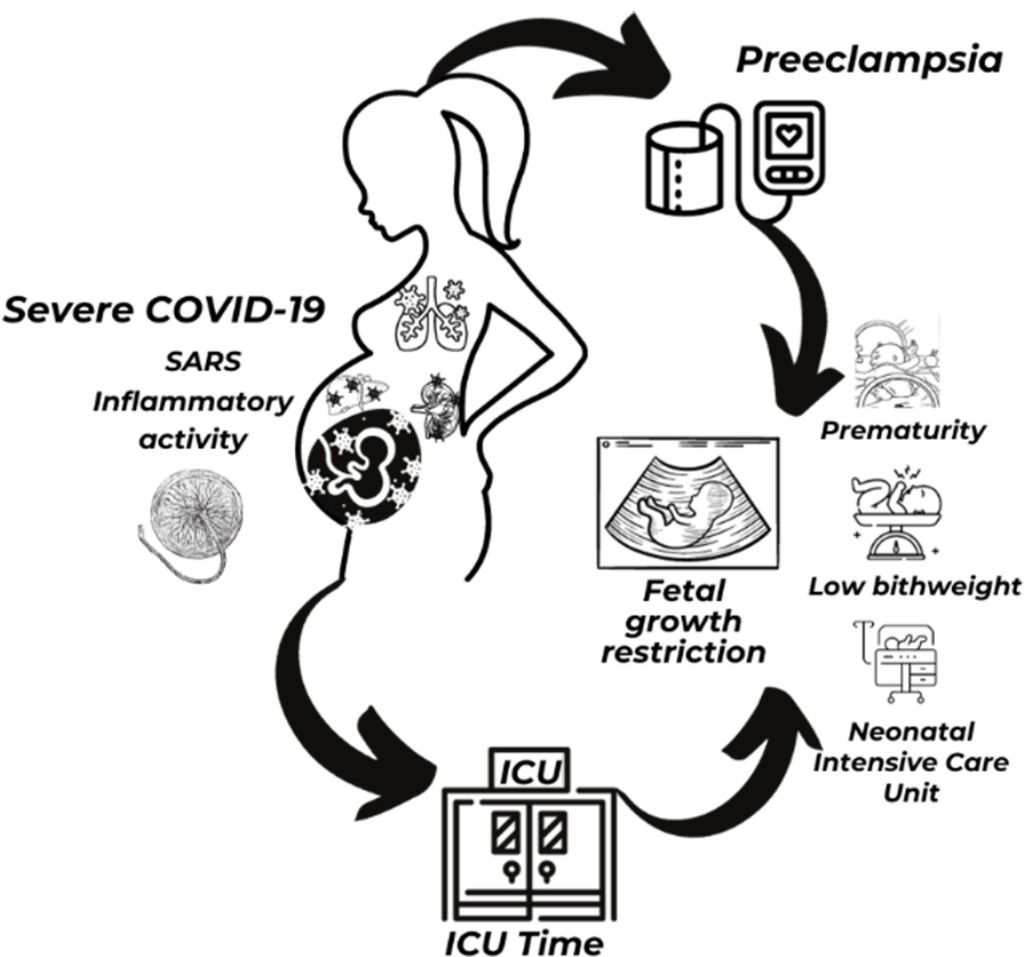

**Figure 5** **Illustration of COVID-19, preeclampsia, maternal and perinatal outcomes.** Notes: ICU, intensive care unit; SARS, severe acute respiratory syndrome. Figure created in Canva Pro. https://www.canva.com/.

HELLP syndrome. The universal screening for SARS-CoV-2 was not performed, and the asymptomatic patients were not included. Type 2 error may have occurred due to the sampling imbalance between the PE+ and PE- groups, which must be interpreted with due caution.

## CONCLUSIONS

When COVID-19 is associated with pregnancy, it increases the prevalence of PE by 11%. This is about 3.7 times higher than data from the pre-pandemic period, in the same institutions. Severe COVID-19 was the main predictive factor for PE, increasing its risk by eight times, and fetal growth restriction was the main perinatal outcome of PE, with the same magnitude. Pregnant women with PE were two times more likely to require ICU admission, and the length of stay in the ICU was longer in PE+ group. Associated comorbidities were more frequent in the PE+ group. Maternal mortality was more frequent

among pregnant women with PE, but there was no significant difference between groups. The placental morphological changes were 30% higher in the PE+ group. Prematurity, low birth weight and need for NICU admission were also more prevalent in the PE+ group.

## ACKNOWLEDGEMENTS

Our thanks to the Academic Publishing Advisory Center (CAPA) teams of the Federal University of Paraná (UFPR) for working on translating our manuscript into English.

### Funding
The APC has been supported by the FUNPAR Foundation, Ministry of Education—Brazil and Federal University of Paraná. There was no additional external funding received for this study. The funders had no role in study design, data collection and analysis, decision to publish, or preparation of the manuscript.

### Grant Disclosures
The following grant information was disclosed by the authors:
The FUNPAR Foundation, Ministry of Education—Brazil and Federal University of Paraná.

### Competing Interests
The authors declare there are no competing interests.

### Author Contributions

- Narcizo L.E.C. Sobieray conceived and designed the experiments, performed the experiments, analyzed the data, prepared figures and/or tables, authored or reviewed drafts of the article, and approved the final draft.
- Newton S. Carvalho conceived and designed the experiments, performed the experiments, analyzed the data, prepared figures and/or tables, authored or reviewed drafts of the article, and approved the final draft.
- Cynthia F. Klas analyzed the data, prepared figures and/or tables, authored or reviewed drafts of the article, and approved the final draft.
- Isabella N. Furuie conceived and designed the experiments, performed the experiments, authored or reviewed drafts of the article, and approved the final draft.
- Jullie A. Chiste conceived and designed the experiments, performed the experiments, authored or reviewed drafts of the article, and approved the final draft.
- Cyllian A. Fugaça conceived and designed the experiments, performed the experiments, authored or reviewed drafts of the article, and approved the final draft.
- Jessica S. Longo conceived and designed the experiments, performed the experiments, authored or reviewed drafts of the article, and approved the final draft.
- Juliana D. Oliveira performed the experiments, authored or reviewed drafts of the article, and approved the final draft.
- Sérgio L. Padilha conceived and designed the experiments, analyzed the data, prepared figures and/or tables, and approved the final draft.

## Human Ethics

The following information was supplied relating to ethical approvals (i.e., approving body and any reference numbers):

Human Research Ethics Committee of the Hospital de Clínicas Complex of the Federal University of Paraná

## Data Availability

The raw measurements are available in the Supplementary File.

## Supplemental Information

Supplemental information for this article can be found online at http://dx.doi.org/10.7717/peerj.17481#supplemental-information.

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
