# Peer review of "Preeclampsia in pregnant women with COVID-19: a prospective cohort study from two tertiary hospitals in Southern Brazil"

_PeerJ, doi:10.7717/peerj.17481_

## Round 0.1 · original submission · Minor Revisions

· Academic Editor

Minor Revisions

Please refer to ALL comments submitted by reviewers.

Reviewer 1 ·

Basic reporting

### Data
Based on the raw data, 1 indicates the presence of the condition, 0 indicates absence. This is not consistent with Nulliparity – there are only two patients with the condition (according to the raw data), but Table 1 shows 9.
Variable “ICU length of stay (days) >7”: table 3 says 5, raw data contains 4 – which one is it? Refer back to the “Article structure” part – is this an interval and if so – the data was not shared…
Variable “Maternal corticoid therapy”: shown in Table 3 but not present in the raw data provided.
### Article structure
Figure 2 – the note explains what OR stands for, but the figure does not have it. The X-axis could be marked with OR (like it was done in Figure 3).
Separation of decimals is not consistent throughout the text and Table 3 – both “.” and “,” are used…
Table 3 does not indicate whether mean or median was used (like Table 2 does, e.g.).
Table 3 uses both whole numbers (e.g. 6) and decimal numbers (e.g. 7,0) to represent n. Should be consistent (like in Tables 1 and 2).
Table 3 should only represent a number (n) with percentage (%) but it also contains intervals – this is unclear.
### Language
The article needs additional proofreading. E.g.:
- Line 29-32: The sentence should start as: “The main objectives of this study were to estimate the prevalence…” ; overall this sentence is too long/confusing – dividing it would make it more readable. Additionally, the end of the sentence can be improved, e.g. “… in hospitalized pregnant women with COVID-19 and preeclampsia compared to those without preeclampsia.”
- “Similarly” is used a lot in the text when “similar” should have been used.
- Line 54: “admission in Intensive Care Unit” should be “admission to Intensive Care Unit”
- Line 67: “among pregnancy” should be changed to “in pregnancy”
- Line 225-226: “On the other hand, when in the pregnant woman with COVID-19 the PE is installing the chance of the severity of the case and the necessity of ICU has increased (Figure 5).” – the authors probably meant “increasing”, not “installing”
- Line 230: “The universal screening to SARS-CoV-2, was not performed, “… to should probably be replaced with for, and the first comma is not necessary.
Note: this list is not exhaustive, it only contains some examples – the article should be proofread.
### References
The authors conclude that Severe COVID-19 was the main risk factor for preeclampsia… However, they miss the opportunity to compare their results to the studies that have had (somewhat) different conclusions to theirs. E.g.
https://www.ncbi.nlm.nih.gov/pmc/articles/PMC8480209/
https://www.sciencedirect.com/science/article/pii/S2210778921003214
https://www.sciencedirect.com/science/article/pii/S246878472200143X
https://www.sciencedirect.com/science/article/pii/S2210778922000484

Experimental design

Overall well designed and executed. However, one point is not completely clear, regarding sample size estimation. Have the authors taken into consideration the large difference in number of patients between groups? Which formula was used to estimate the sample size?

Validity of the findings

No major comments here except what has been mentioned in the “Basic Reporting” part. The authors should refer to the comments there and implement the changes or explain further.

Additional comments

No comment.

Reviewer 2 ·

Basic reporting

the manuscript is interesting and generally well written.

Please, improve the quality of figure 1 and 5

Table 1: gestational age at time of COVID-19 test and delivery must be added

Line 53-56: a more precise introduction on COVID-19 and SARS-CoV-2 is needed. In fact, COVID-19 is mainly know as a respiratory disease. However, it deserves to be specified that SARS-CoV-2 infection can also cause non-respiratory complications (see PMID: 35114008, PMID: 38543856). This is an important point to add since it highlights the complexity of this pathology.

Lines 59-62: references must be added

Lines 61-62: it deserves to be pointed out that the inappropriate trophoblastic invasion of the spiral arterioles of the uterus leads to an hypoxic environment, oxidative stress and, local and systemic inflammation (see PMID: 37296665). All these are characteristics of PE pregnancies.

Experimental design

Authors must define what are the symptoms and charactheristics that allow the definition of severe COVID-19 since it has also been reported a mild and asymptomatic form of the disease which, in some cases, also caused PE onset (as reviewed here PMID: 35943095)

Validity of the findings

All underlying data have been provided.

---

## Round 0.2 · accepted · Accept

· Academic Editor

Accept

Dear Authors,

You have addressed all comments raised by reviewers.

Reviewer 2 ·

Basic reporting

improved

Experimental design

improved

Validity of the findings

improved

Additional comments

the manuscript has been improved and can be accepted in the present form